# Resveratrol attenuates non-steroidal anti-inflammatory drug-induced intestinal injury in rats in a high-altitude hypoxic environment by modulating the TLR4/NFκB/ IκB pathway and gut microbiota composition

**Shenglong Xue[1], Wenhui Shi[2,3], Tian Shi[4,5], Ailifeire Tuerxuntayi[6], Paziliya Abulaiti[6], Zhuoshuyi Liu[6], Najimangu Remutula[6], Kailibinuer Nuermaimaiti[6], Yingying Xing[6], Kudelaiti Abdukelimu[6], Weidong Liu[1,4], Feng Gao[4,5]***

1 College of Life Science and Technology, Xinjiang University, Urumqi, China, 2 General Hospital of Xinjiang Military Region of PLA, Urumqi, China, 3 Key Laboratory of Special Environmental Medicine of Xinjiang, Urumqi, China, 4 Department of Gastroenterology, People's Hospital of Xinjiang Uygur Autonomous Region, Urumqi, China, 5 Xinjiang Clinical Research Center for Digestive Diseases, Urumqi, China, 6 Xinjiang Medical University, Urumqi, China

* xjgf@sina.com

## Abstract

## Introduction

Non-steroidal anti-inflammatory drugs (NSAIDs) are currently the most widely used anti-inflammatory medications, but their long-term use can cause damage to the gastrointestinal tract(GIT). One of the risk factors for GIT injury is exposure to a high-altitude hypoxic environment, which can lead to damage to the intestinal mucosal barrier. Taking NSAIDs in a high-altitude hypoxic environment can exacerbate GIT injury and impact gut microbiota. The aim of this study is to investigate the mechanisms by which resveratrol (RSV) intervention alleviates NSAID-induced intestinal injury in a high-altitude hypoxic environment, as well as its role in regulating gut microbiota.

## Methods

Aspirin was administered orally to rats to construct a rat model of intestinal injury induced by NSAIDs. Following the induction of intestinal injury, rats were administered RSV by gavage, and the expression levels of TLR4, NF-κB,IκB as well as Zonula Occludens-1 (ZO-1) and Occludin proteins in the different treatment groups were assessed via Western blot. Furthermore, the expression of the inflammatory factors IL-10, IL-1β, and TNF-α was evaluated using Elisa.16sRNA sequencing was employed to investigate alterations in the gut microbiota.

**Data Availability Statement:** All relevant data are within the manuscript and its Supporting Information files.

**Funding:** This research was funded by National Natural Science Foundation of China, grant number 82260116; Natural Science Foundation of Xinjiang Uygur Autonomous Region, grant number ZYYD2022A06.

## Results

The HCk group showed elevated expression of TLR4/NF-κB/IκB pathway proteins, increased expression of pro-inflammatory factors IL-1β and TNF-α, decreased expression of the anti-inflammatory factor IL-10, and expression of intestinal mucosal barrier proteins ZO-1 and Occludin. The administration of NSAIDs drugs in the plateau hypoxic environment exacerbates intestinal inflammation and damage to the intestinal mucosal barrier. After treatment with RSV intervention, the expression of TLR4/NF-κB/IκB signaling pathway proteins would be reduced, thereby lowering the expression of inflammatory factors in the HAsp group. The results of HE staining directly show the damage to the intestines and the repair of intestinal mucosa after RSV intervention. 16sRNA sequencing results show significant differences (P<0.05) in *Ruminococcus*, *Facklamia*, *Parasutterella*, *Jeotgalicoccus*, *Coprococcus*, and *Psychrobacter* between the HCk group and the Ck group. Compared to the HCk group, the HAsp group shows significant differences (*P*<0.05) in *Facklamia*, *Jeotgalicoccus*, *Roseburia*, *Psychrobacter*, and *Alloprevotella*. After RSV intervention, *Clostridium_sensu_stricto* bacteria significantly increase compared to the HAsp group.

## Conclusion

Resveratrol can attenuate intestinal damage caused by the administration of NSAIDs at high altitude in hypoxic environments by modulating the TLR4/NF-κB/IκB signaling pathway and gut microbiota composition.

## 1. Background

Nonsteroidal anti-inflammatory drugs (NSAIDs) are a class of anti-inflammatory drugs that do not contain glucocorticoids and are currently the most widely used. They have anti-inflammatory, anti rheumatic, analgesic, and anticoagulant effects, and can effectively alleviate pain and inflammation [1]. However, multiple studies have shown that long-term use of NSAIDs can cause intestinal damage, disrupt the intestinal mucosal barrier [2–4], and lead to the occurrence of NSAIDs related intestinal diseases. Among patients with intestinal diseases caused by NSAIDs, some suffer from severe intestinal symptoms and even concurrent small intestine bleeding, perforation, ulcers, etc [5].

The plateau environment is a critical habitat for human survival, marked by low pressure, hypoxia, and intense ultraviolet radiation. The high-altitude hypoxic conditions can result in detrimental effects on various organs such as the heart, brain, and lungs. Some studies have shown that plateau hypoxia may contribute to the development of digestive disorders, and that exposure to plateau hypoxia severely damages the intestinal mucosal structure and intestinal microecology, thereby causing damage to the intestinal tract and the intestinal mucosal barrier and altering the composition of intestinal microorganisms [6–8]. Additionally, high-altitude hypoxic environments can incite intestinal inflammation, activating the NF-κB pathway in the intestinal tissues. This activation, involving NF- κB as a transcription factor and interferon regulatory factor, augments the production of various inflammatory factors, ultimately inducing intestinal inflammation [9, 10]. In an animal study, it was found that the expression of pro-inflammatory cytokines could be decreased and the expression of anti-inflammatory cytokines could be increased by inhibiting the activation of the TLR4/NF-κB signaling pathway,

resulting in the reduction of intestinal inflammation [11]. The hypoxic environment activates the TLR4/NF-κB signaling pathway in the intestinal tissues, which is one of the contributing factors to the frequent occurrence of various intestinal diseases, including inflammatory bowel disease (IBD), in the plateau population [12]. With more than 30 million individuals worldwide consuming NSAIDs daily, their use in high-altitude hypoxic regions may exacerbate intestinal damage. Despite the widespread attention to the long-term use of NSAIDs and their effects on small intestine damage, there remains a gap in research exploring the impact of NSAIDs on intestinal damage in populations residing in high-altitude hypoxic environments. Considering the compounded stressors in high-altitude settings and NSAID usage, the administration of NSAIDs in high-altitude hypoxic environments may further intensify intestinal damage. Consequently, the intervention and treatment of NSAIDs also represent a significant research focus.

Resveratrol (RSV), a polyphenolic organic compound, exhibits antioxidant, anti-inflammatory, antiviral, anticancer, and antiaging properties. It is mainly extracted from plants such as Tiger Balm, Tiger Balm Cassia, and Grape [13]. RSV, being a compound molecule extracted from a natural product, has been extensively studied. Its multifaceted properties make it effective in treating various diseases with therapeutic effects. It has been demonstrated that RSV has a protective effect on the intestinal tract and is effective in treating various intestinal disorders, such as inflammatory bowel disease and colitis [14]. A systematic review by Sandoval-Ramírez et al. describes the effects of phenolic compounds on the intestinal barrier and finds that oral administration of phenolic chemistry to animals improves the integrity and function of the intestinal barrier, improves the expression of tight junction proteins and enhances intracellular antioxidant activity. It is suggested that phenolic compounds may be used for the treatment of intestinal damage in humans, especially resveratrol, which is the most studied [15]. In addition, in animal experiments, it has also been found that RSV can effectively alleviate intestinal inflammation, maintain the integrity of the intestinal mucosal barrier, and reduce intestinal injury by modulating the TLR4/NF-κB signaling pathway [16]. Therefore, RSV plays a role as a good preventive drug in preventing inflammation and protecting the intestinal barrier.

The aim of this study was to replicate the plateau hypoxia environment and orally administer aspirin to rats. We investigated the effects of NSAID administration on intestinal damage in rats under plateau hypoxia, and further examined the potential of Rsv to alleviate intestinal inflammation caused by NSAIDs through the TLR4/NF-κB/IκB signaling pathway. Furthermore, we analyzed the changes in the composition of rat intestinal flora under different treatment conditions using 16sRNA sequencing.

## 2. Material and methods

### 2.1. Reagents and drugs

Aspirin was purchased from Bayer Healthcare GmbH, Germany. Resveratrol was purchased from Sigma Company, USA, with 98% purity, commodity number: V900386. pentobarbital sodium was purchased from Sinopharm Chemical Reagent Co., Ltd.

### 2.2. Experimental animals and environment

This experiment was approved by the Animal Ethics Committee for Research and Testing of Xinjiang Medical University (Approval number: KY20230209135). It was also conducted in accordance with the Guidelines for Ethical Review of Animal Welfare (GB/T 35892–2018). The test animals were housed in the Northwest Key Laboratory of Special Environmental Medicine, located at the General Hospital of Xinjiang Military Region of the Armed Police in Xinjiang, China. Seventy male SD rats of SPF grade were utilized as test animals, procured

from Xinjiang Medical University, located in Xinjiang, China. All rats had an initial body weight ranging between 160-180g. The rats were pre-housed in an SPF-grade animal laboratory, maintaining a room temperature of 20±2˚C and a humidity of (45±5)% RH, and provided ad libitum access to water and food to facilitate acclimatization. After one week of pre-feeding, the rats were placed in a plateau hypoxia simulation chamber to replicate a hypoxic environment resembling an altitude of 5500 m and atmospheric pressure of 379 mmHg. The test rats were housed in the plateau hypoxic environment, with the exception of the gavage period (12:30–1:30 every day). All efforts were made to minimize their suffering and all experiments were conducted in accordance with the ARRIVE guidelines.

## 2.3. Experimental design

70 rats were randomly divided into 7 groups, each containing 10 rats, and were subsequently numbered based on their respective groups. The rats were categorized into injury simulation groups: the plain blank group (saline gavage at 100 m above sea level), the plain NSAID-treated group (gavage with 200 mg/kg of aspirin at 100 m above sea level), the plateau blank group (saline gavage after entering a simulated low-pressure oxygen chamber), and the plateau NSAID-treated group (gavage with 200 mg/kg of aspirin after entering a simulated low-pressure oxygen chamber). Additionally, the resveratrol treatment group was subjected to simulated low-pressure oxygen chamber exposure and received 200mg/kg aspirin via gastric gavage, followed by resveratrol administration 30 minutes later. Resveratrol was administered in low-dose (25mg/kg), medium-dose (50mg/kg), and high-dose (100mg/kg) groups.

The gavage treatments outlined above were administered for a duration of three weeks. After the final gavage, the rats were provided with water but no food, weighed after 12 hours, anesthetized intraperitoneally with 2% pentobarbital sodium, and subsequently euthanized by collecting blood from the orbital sinus and performing cervical dislocation. Subsequently, the all rats underwent dissection, during which their intestinal tissues were excised, rinsed with saline to eliminate blood contamination, and preserved in LPS solution for multiple subsequent biochemical and histopathological analyses.

## 2.4. Rat jejunal injury score

After execution, the rats were dissected, and the abdominal cavity was opened to observe the abdominal cavity. After taking the proximal jejunum 2 cm and rinsing it with ice saline, the specimen was laid flat on a cardboard with the intestinal mucosa layer upward and fixed with a big head pin to observe the degree of intestinal mucosal damage. The severity of small intestine damage in the rats was scored broadly by two specialists using a blinded method based on the Reuter Scale [17] (S1 Table).

## 2.5. HE staining

The jejunal specimens were fixed in 10% neutral formaldehyde solution, followed by dehydration and paraffin embedding. Subsequently, 5-μm-thick sections were prepared for morphological evaluation. The sections were then dewaxed in xylene and sequentially treated with a series of graded alcohols for subsequent hematoxylin-eosin (HE) staining. Finally, observe the changes in the intestinal mucosa under a light microscope by two pathology specialists using a blinded method and the intestinal damage was scored according to Chiu's scale [18, 19] (S2 Table).

## 2.6. Enzyme-linked immunosorbent assay

The levels of TNF-α, IL1β, and IL-10 in serum, as well as TNF-α, IL1β, IL-10, SOD, and MPO in jejunal tissues, were assessed using enzyme-linked immunosorbent assay (ELISA) kits, following the manufacturer's protocols (Weiao bio, Shanghai). More specifically, serum samples and enzyme labeling reagents were applied to 96-well polystyrene microtiter plates that were pre-coated with antibodies targeting TNF-α, IL1β, IL-10, SOD, and MPO. Following incubation at 37°C for 60 minutes, the samples were subjected to TMB color development solution, terminated with a termination solution, and their absorbance values were subsequently measured at 450 nm.

## 2.7. Western blotting

Inflammatory pathway proteins, including TLR4, NF-κB, and IκB, along with intestinal mucosal barrier proteins ZO-1 and Occludin, were analyzed using Western blotting. The jejunal samples were homogenized in RIPA lysis buffer supplemented with protease inhibitors, followed by the collection of the supernatant after centrifugation and subsequent determination of the protein concentration. Equal amounts of total proteins were separated by SDS-PAGE, transferred to polyvinylidene difluoride (PVDF) membranes, and the latter were then sealed with 5% skim milk for 2 hours. The membranes were incubated with primary antibodies overnight at 4°C, followed by four rinses with TBST. The integrated density value of protein bands was quantified using Image J software, with GAPDH/β-actin serving as the internal reference. For the primary antibodies, the sources and concentrations were as follows: Occludin (1:1000 rabbit source), ZO-1 (1:500 rabbit source), TLR4 (1:500 rabbit source), IκB (1:1000 rabbit source), NF-κB (1:500 rabbit source), β-actin (1:2000 mouse source). The secondary antibodies used were sheep anti-rabbit secondary antibody (Weiao 1:2000) and sheep anti-mouse secondary antibody (Weiao 1:2000).

## 2.8. 16sRNA sequencing for gut microbial analysis

Six rats were taken from each group and their last feces before execution were taken for intestinal microbial analysis. Flora DNA was extracted from the rat fecal samples using the TopTaq DNA Polymerase kit from Transgen (China). DNA quantification was performed using a NanoDrop 2000 spectrophotometer (Thermo Scientific, Wilmington, NC, USA), followed by additional evaluation using a 1% agarose gel. Amplification of the V3-V4 highly variable region of the 16S rRNA gene was performed using specific primer pairs with barcodes (forward primer: 5'−CCTACGGGGNGGCWGCAG−3';reverse primer: 5'−GACTACHVGGGTAT CTAATCC−3'). Each sample was independently amplified three times. Finally, the PCR products were detected using agarose gel electrophoresis and then pooled together from the corresponding sample. The pooled PCR products were utilized as templates for index PCR, in which index primers were used to incorporate the Illumina index into the library. The amplification products were then detected by gel electrophoresis and purified using the Agencourt AMPure XP Kit from Beckman Coulter (CA, USA). The purified products were then cataloged into 16S V3-V4 libraries. The quality of the libraries was assessed using a Qubit@2.0 Fluorometer from Thermo Scientific and an Agilent Bioanalyzer 2100 system. Finally, the pooled libraries were sequenced using an Illumina MiSeq 250 sequencer, resulting in the generation of 2 × 250 bp paired-end reads.

## 2.9. Data statistics and analysis

Data were processed using the SPSS 26.0 software. The data were expressed as mean ± standard deviation (SD). Multiple comparisons between the groups were conducted by one-way analysis of variance (ANOVA) followed by the LSD post hoc test. $p < 0.05$ was considered as significantly different. Graphing was done through Graphpad prism 8.0 software. Univariate and bivariate correlation analysis was performed through R Studio. The * in this study represents comparisons with the plains blank group:*$P<0.05$, **$P<0.01$, ***$P<0.001$, # Comparisons with the plateau NSAIDs-treated group: #$P<0.05$, ##$P<0.01$, ###$P<0.001$.

# 3. Results

## 3.1 Jejunal injury score

At the time of dissection, tissues from various intestinal sites of each rat were preserved and assessed for tissue damage in distinct segments of the rat intestine using the Reuter scale. In this investigation, the jejunum was selected as the primary focus for tissue damage assessment. The analysis revealed a significantly higher degree of jejunum injury ($P < 0.05$) in the PAsp group compared to the Ck group, suggesting a correlation between Aspirin gavage and intestinal injury in rats. Furthermore, rats in the HAsp group exhibited a higher degree of jejunal injury compared to those in the HCk group, implying that Aspirin administration in the hypoxic plateau environment exacerbates intestinal injury. Following intervention with RSV, varying degrees of improvement were observed in jejunal injury, with a significant reduction in intestinal injury noted in rats from the RSVM group ($P < 0.05$) (Fig 1).

## 3.2 HE staining results

The histopathological results of the jejunum in various treatment groups were observed using HE staining, as depicted in Fig 2. In the Ck group, the small intestinal mucosal villi appeared normal, with tightly arranged cells and no evident pathological changes. In the PAsp group, signs of atrophy were observed in the intestinal mucosal villi, along with a few necrotic

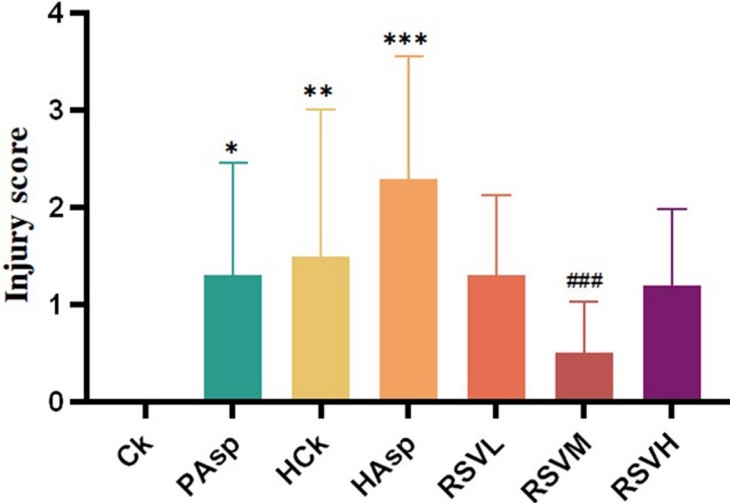

**Fig 1. Rat jejunal tissue damage score.** Comparison with plain blank group:*$P<0.05$, **$P<0.01$, ***$P<0.001$; Comparison with plateau NSAIDs: #$P<0.05$, ##$P<0.01$, ###$P<0.001$.

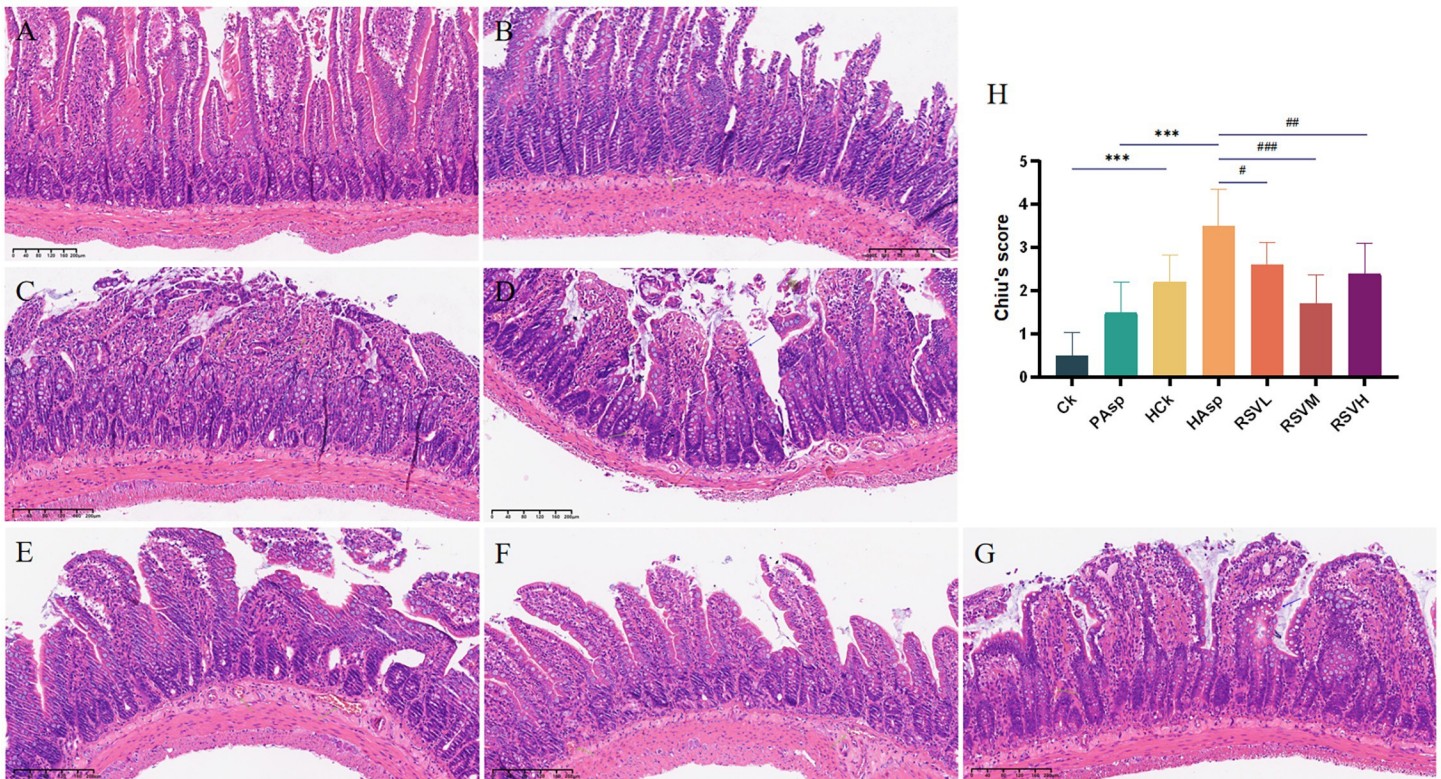

**Fig 2. Histopathological section results of rat jejunum.** (A) plain blank group (Ck); (B) plain aspirin-treated group (PAsp); (C) plateau blank group (HCk); (D) plateau aspirin-treated group (HAsp); (E) low-dose resveratrol-treated group (RSVL); (F) medium-dose resveratrol-treated group (RSVM); (G) high-dose resveratrol-treated group (RSVH); (H) Chiu's of the jejunal tissue score.

epithelial cells. The HCk group exhibited atrophied intestinal mucosal villi, with some villi partially detached, accompanied by multiple hemorrhagic dots and inflammatory cell infiltration in the pathology section. In the HAsp group, the intestinal mucosal villi were severely atrophied with large areas of detachment, the lamina propria was edematous, and there was an increase in lymphocytes containing multiple hemorrhages. Following RSV intervention, a significant reduction in intestinal damage was observed ($P < 0.05$), accompanied by the restoration of intestinal villi to their normal state. This suggests that RSV effectively alleviates intestinal damage induced by aspirin administration under plateau hypoxic conditions.

## 3.3 Results of oxidative stress indicators

Oxidative stress indicators are believed to contribute to the initiation and perpetuation of intestinal inflammation [20]. We assessed the expression of oxidative stress indicators Myeloperoxidase (MPO) and Superoxide dismutase (SOD) in blood using ELISA (Fig 3). Aspirin administration was observed to elevate MPO expression and diminish SOD expression ($P < 0.05$). Following aspirin administration under plateau hypoxia, MPO expression was found to be further increased, while SOD expression was further decreased compared to the PAsp group. This implies that the decreased oxygen content in the plateau hypoxic environment might prompt the body to upregulate MPO production and downregulate SOD production. Aspirin administration to rats in this environment exacerbates oxidative stress levels. Subsequent to intervention with RSV, a significant reduction in MPO expression level and an increase in SOD expression level were observed, indicating the potential of RSV intervention

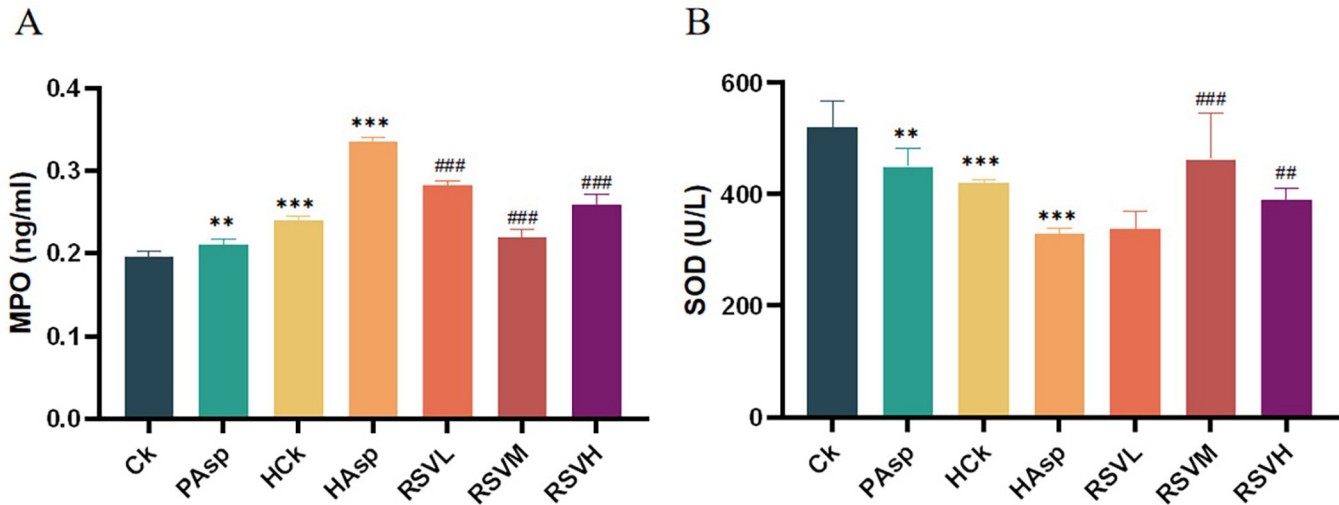

**Fig 3. Indicators of oxidative stress.** (A) Changes in myeloperoxidase in different treatment groups; (B) Changes in superoxide dismutase in different treatment groups.

to effectively mitigate oxidative stress in rats($P < 0.05$). Among the three RSV-treated groups with varying doses, the RSVM group exhibited the most favorable outcomes, indicating that administering RSV at 50 mg/kg might be optimal for intervention treatment.

## 3.4. Results of Elisa

We conducted a comprehensive examination of the expression levels of the pro-inflammatory cytokines IL-1β, IL-10 and TNF-α in the jejunum and serum of rats using the enzyme-linked immunosorbent assay (ELISA). The expression levels of IL-1β and TNF-α in the jejunum and blood were found to be significantly higher than those in the control group after administration of Aspirin via gavage to rats, while the opposite was true for IL-10. Furthermore, administering Aspirin via gavage to rats under plateau hypoxia conditions would further increase the release of pro-inflammatory factors and decrease the release of anti-inflammatory factors. This suggests that administering Aspirin under plateau hypoxic conditions exacerbates intestinal inflammation and may lead to intestinal injury. In addition, our findings indicate that RSV intervention significantly decreased the expression of pro-inflammatory factors IL-1β and TNF-α and increased the expression of IL-10 in both blood and jejunum ($P < 0.05$). These results suggest that RSV is effective in attenuating intestinal damage caused by the administration of NSAIDs under hypoxic conditions at high altitudes. Additionally, after intervention with three different doses of RSV, the RSVM group (50 mg/kg) exhibited the most effective relief of intestinal inflammation (Fig 4).

## 3.5. Western blot results

The expression levels of intestinal mucosal barrier proteins Occludin and ZO-1, as well as the inflammatory pathway proteins IKB, TLR4, and NF-κB, were examined using Western Blot (WB). Occludin and ZO-1 are crucial constituents of intercellular junctions, playing pivotal roles in maintaining and regulating the structural integrity of epithelial cell junctions. The interaction between ZO-1 and Occludin proteins is vital for both the formation and functionality of intercellular junctions; any aberrations in their expression or mutations can result in disruption of the intestinal epithelial barrier. Our findings indicate that Aspirin administration

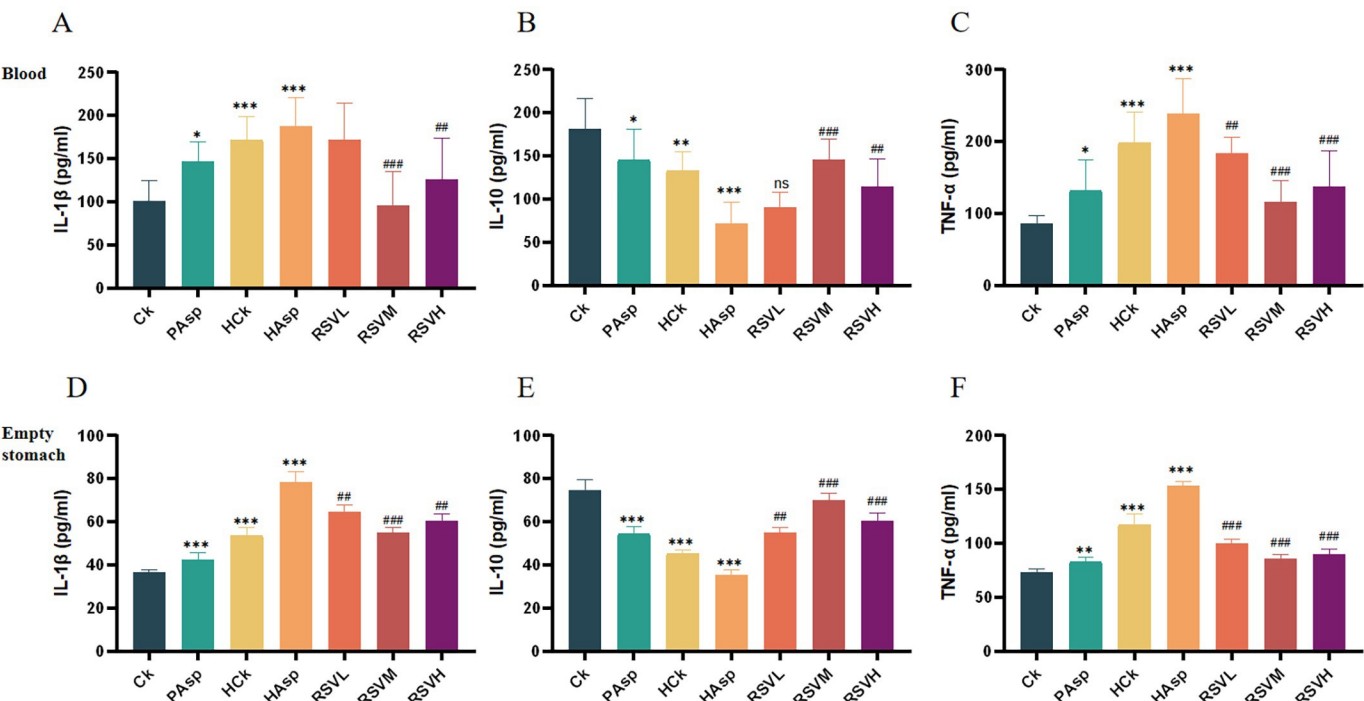

**Fig 4. Expression of inflammatory factors in jejunum and blood.** (A) Blood IL-1β; (B) Blood IL-10; (C) Blood TNF-α; (D) Jejunal IL-1β; (E) Jejunal IL-10; (F) Jejunal TNF-α.

via gavage decreased the expression of ZO-1 and Occludin proteins in rats. Furthermore, in a plateau hypoxic environment, Aspirin gavage exacerbated this reduction, potentially intensifying the disruption of the intestinal barrier. Following RSV intervention, a substantial increase in the expression levels of ZO-1 and Occludin proteins was observed (Fig 5A–5C), suggesting the potential of RSV to mitigate intestinal mucosal injury induced by Aspirin gavage in rats exposed to a plateau hypoxic environment. Additionally, we assessed the expression levels of TLR4, NF-κB, and IκB proteins under various treatment conditions to explore the potential of RSV in mitigating intestinal inflammation and injury through the regulation of the TLR4/NF-κB/IκB signaling pathway, thereby reducing associated inflammatory factors. The results demonstrated that RSV significantly decreased the expression levels of TLR4 and NF-κB proteins while increasing the expression level of IκB protein (Fig 5A and 5D–5F).

## 4. 16sRNA sequencing results

### 4.1. Number of OTUs

By analyzing the microbial communities of the different groups of treatments, it was found that there was variability in the distribution of microbial species among the different treatment groups. The Venn diagram presented in Fig 6A illustrates the OTU counts for each treatment group: Ck (1226 OTUs), PAsp (1102 OTUs), HCk (1537 OTUs), HAsp (968 OTUs), RSVL (1060 OTUs), RSVM (812 OTUs), and RSVH (956 OTUs). The total number of OTUs across all seven groups was 313. These findings suggest that both Aspirin and RSV treatments induced alterations in the composition of rat gut microbes (Fig 6A).

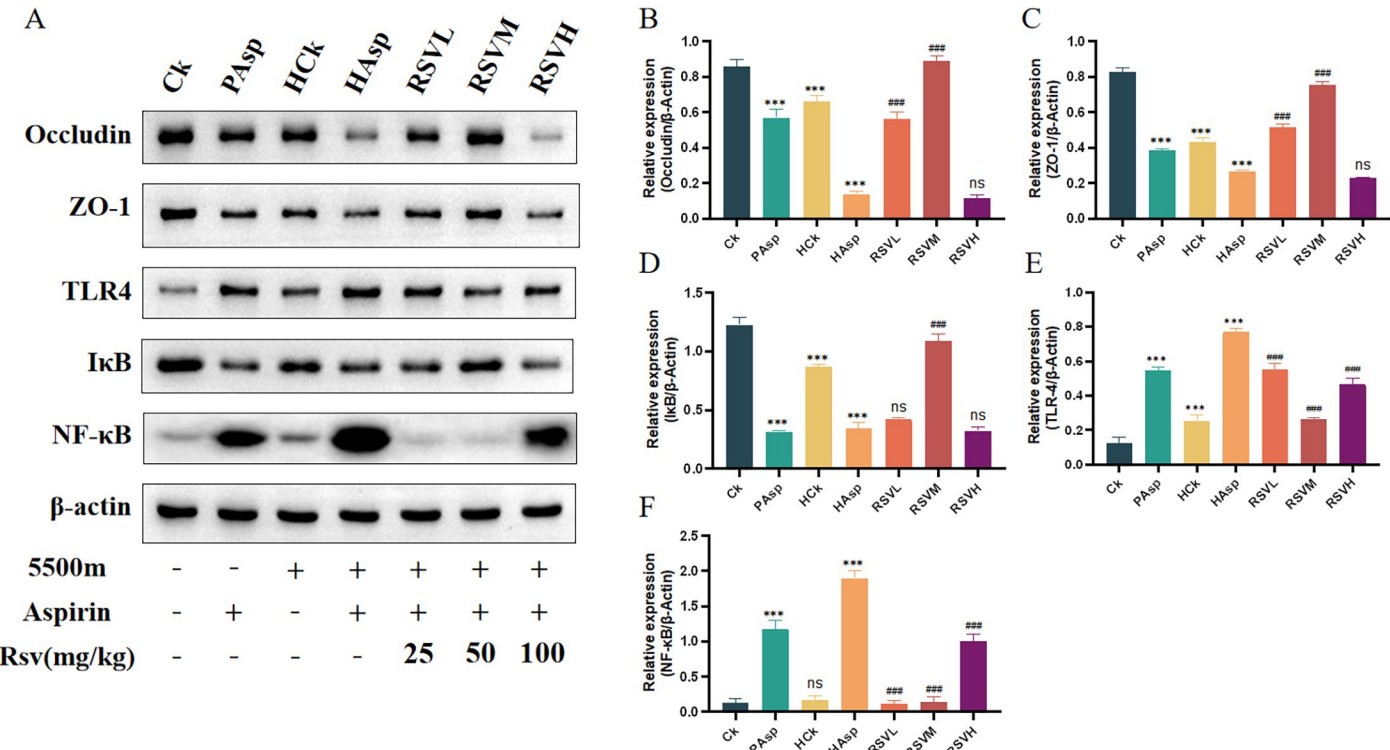

**Fig 5. Resveratrol intervention inhibits TLR4/NF-κB/IκB-mediated inflammation.** (A) WB blotting results of Occludin, ZO-1, TLR4, IκB, NF-κB, and β-actin; (B) Illustration of quantitative Western blotting of Occludin; (C) ZO-1 level; (D) IκB level; (E) TLR4 level; (F) NF- κB level. Band intensities were analyzed with Image J software and normalized with β-Actin.

## 4.2. Bacterial diversity

Analysis of α-diversity and β-diversity indices among the groups revealed differences in both richness and diversity ($P < 0.05$). The Shannon index of the HAsp group was lower than that of the PAsp group, It showed that the HAsp group was less abundant compared to the PAsp group ($P < 0.01$). The diversity of the HAsp group was lower than that of the HCk group, and the abundance of the HAsp group was lower compared to that of the HCk group ($P < 0.05$). These results suggest that Aspirin administration under plateau hypoxia conditions might reduce the species richness of intestinal flora. Following RSV intervention, the differences between the RSVL and RSVH groups were not significant compared to the HAsp group. Additionally, only the Shannon index of the RSVM group was reduced compared to the HAsp group after treatment, suggesting an effect of RSV on the richness of intestinal microbiota in rats. Furthermore, the impact of different dosages of RSV on intestinal microbiota richness varied (Fig 6D).

In the PCoA analysis, the HCk group exhibited a noteworthy level of dispersion and a substantial difference in flora diversity when contrasted with the Ck group($P < 0.05$). The HAsp group showed a greater difference in colony diversity compared to the HCk group ($P < 0.05$). This suggests that both the plateau hypoxic environment and Aspirin contributed to changes in intestinal flora. Following RSV treatment, the degree of dispersion varied among the different dose-treated groups compared to the control group. Specifically, the RSVL and RSVH groups exhibited lower levels of dispersion compared to the HAsp group; furthermore, the difference in flora diversity was not statistically significant. Additionally, the intestinal microflora diversity in rats treated with the RSVM group differed significantly from that of the HAsp

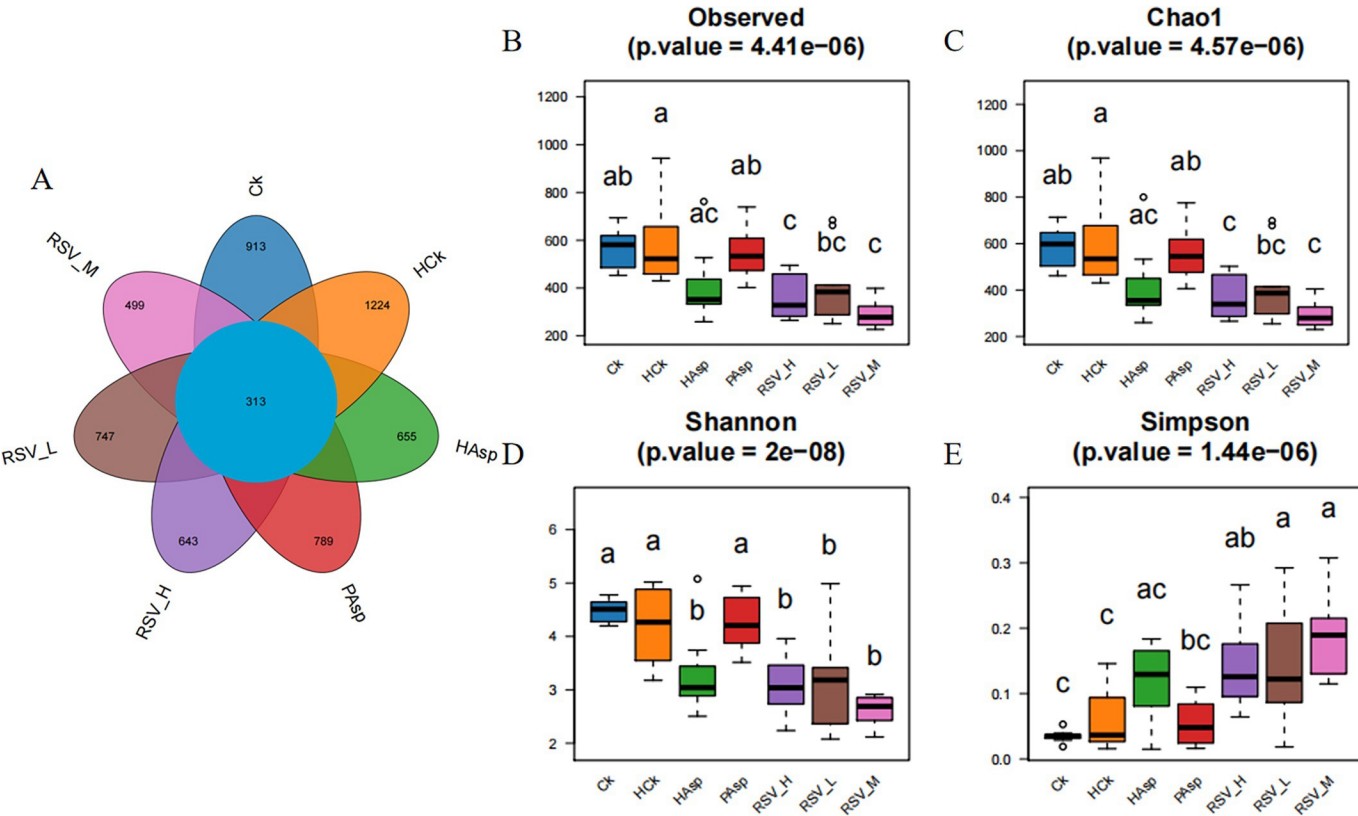

**Fig 6. Venn diagram of OTUs of gut microorganisms with different treatments and Alpha diversity.** (A) Venn diagram of OTUs of gut microorganisms with different treatments; (B) Observed ASVs; (C) Chao1 index; (D) Shannon diversity index; (E) Simpson index.

group. Our findings indicate that plateau hypoxic conditions influenced gut microflora diversity in rats, and the administration of Aspirin could potentially alter the composition of gut microflora diversity in rats. Following intervention by RSVM, the changes in gut flora diversity significantly differed from those observed in the HAsp group. This implies that medium-dose resveratrol might regulate the alterations in gut flora caused by Aspirin administration in rats exposed to plateau hypoxia conditions (S1 Fig).

## 4.3. Microbial composition

Based on the relative abundance, the differential flora of different treatment groups at the phylum and genus levels were identified through ANOVA analysis. Three bacteria showed significant differences at the phylum level: *Firmicutes* ($P < 0.05$), *Bacteroidetes* ($P < 0.05$), and *Actinobacteria* ($P < 0.05$). Both Aspirin and the plateau hypoxic environment increased the abundance of *Firmicutes*: 67.24% in the Ck group, 71.32% in the HCk group, and 84.86% in the HAsp group. The abundance of *Bacteroidetes* decreased: 30.73% in the Ck group, 30.73% in the HCk group, 25.09% in the Ck group, and 13.46% in the HAsp group (Fig 7A).

At the genus level, we identified a total of 35 differential genera. In the HCk group compared to the Ck group, six genera showed differences: *Ruminococcus*, *Facklamia*, *Parasutterella*, *Jeotgalicoccus*, *Coprococcus* and *Psychrobacter*. Five genera differed in the HAsp group compared to the HCk group: *Facklamia*, *Jeotgalicoccus*, *Roseburia*, *Psychrobacter*, and *Alloprevotella*. It is noteworthy that these genera were significantly reduced in the HAsp group compared to the HCk group ($P < 0.05$). These results suggest that aspirin gavage to rats in a

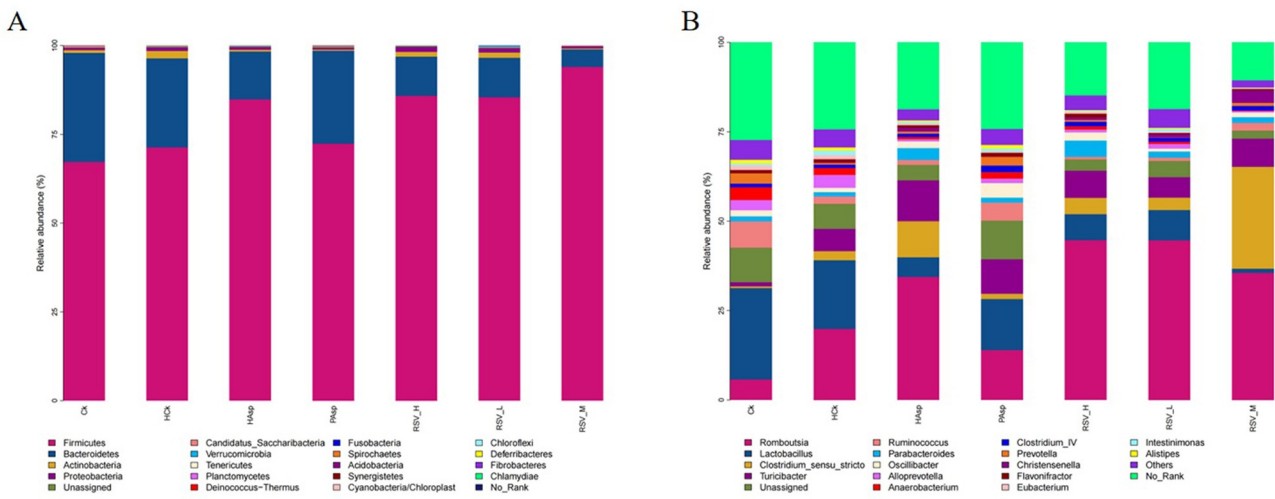

**Fig 7. Bacterial community structure of each group at the phylum level and genus level.** (A) phylum level; (B) genus level.

plateau hypoxic environment affects the composition of rat intestinal flora. Additionally, we found no significant difference in flora between the RSVL and HAsp groups after RSV intervention. *Cupriavidus* was significantly increased in the RSVH group compared to the HAsp group (*P* < 0.05). Two genera differed between the RSVM group and the HAsp group: *Clostridium_sensu_stricto* and *Spirosoma*, both of which were significantly increased in the RSVM group (*P* < 0.05). The above results suggest that RSV intervention also affects the composition of rat intestinal flora (Fig 7B).

## 4.4. Analysis of difference abundance

Differential abundance analysis is utilized to identify taxa exhibiting significant differences in abundance among groups. The LEfSe analysis revealed that the dominant taxa differed across groups: *Bacteroidetes*, *Ruminococcus*, and *Prevotellaceae* in the Ck group; *Alloprevotella*, *Intestinmonas*, and *Roseburia* in the HCk group; *Ruminococcus2*, *Bardyrhizobiaceae*, and *Clostridiales_Incertae_sedis_XI* in the HAsp group; *Ruminococcus_callidus* in the PAsp group; *Peptostreptococcaceae* and *Romboutsia* in the RSV-H group; *Bilophila* in the RSV-L group; and *Clostridiales*, *Clostridia*, and *Clostridium_sensu_stricto* in the RSV-M group. These findings suggest that RSV has the potential to alter the composition and structure of the intestinal flora in rats with Aspirin-induced intestinal injury under plateau hypoxia conditions (Fig 8).

## 4.5. Correlation analysis between inflammatory factors and gut microbiota in mice

To further investigate the interactions between intestinal immune function and intestinal flora in rats across different treatment groups, 35 intestinal bacterial genera with significant differences were subjected to Spearman's rank correlation analysis with the forensic intestinal immune factors IL-10, TNF-α, and IL-1β (Fig 9). The results, depicted in Fig 9, reveal significant negative correlations between the expression levels of *Psychrobacter* and IL-10 (r = -0.75, *P* < 0.05), significant positive correlation between *Psychrobacter* and TNF-α expression level (r = 0.80, *P* < 0.05), and significant negative correlation between *Ruminococcus* and IL-1β expression level (r = -0.82, *P* < 0.05).

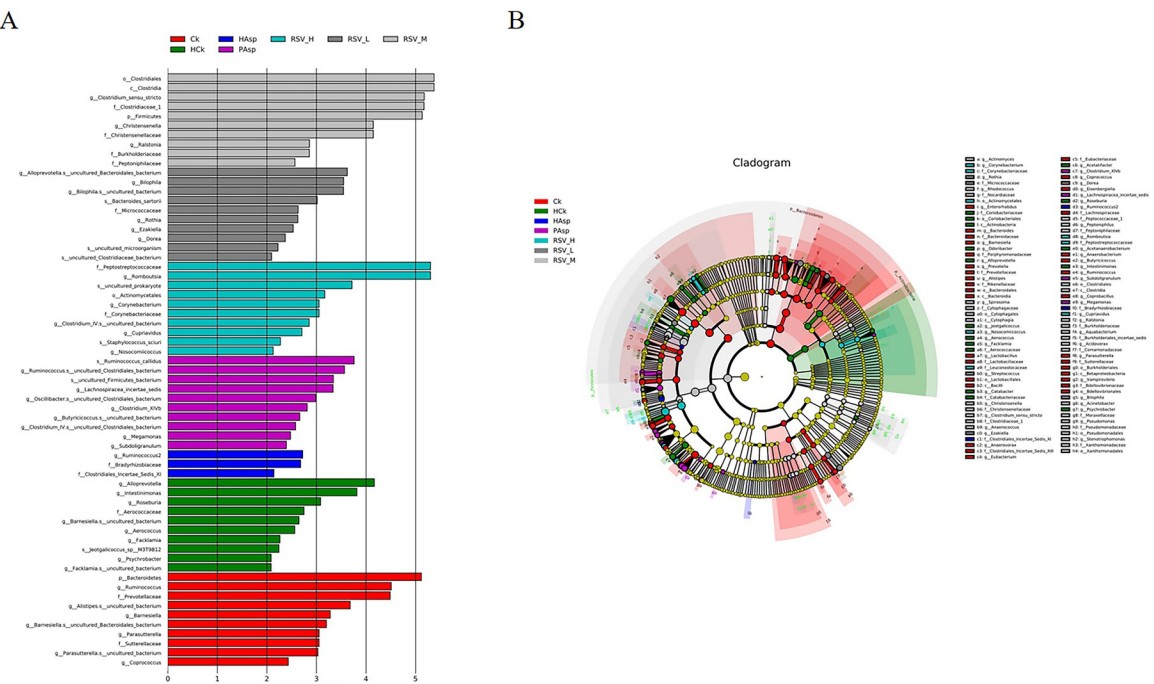

**Fig 8. LEfSe evolutionary branching diagram and the structure of bacterial community of each group at genus level.** (A) LEfSe evolutionary branching diagram, different colors represent different treatment groups:Ck(red), HCk(green), HAsp(blue), PAsp(purple), RSV_H(cyan), RSV_L(dark grey), RSV_M(light grey); (B) Relative abundance of bacterial communities at genus level of different treatment groups.

## 5. Discussion

Plateau hypoxia can induce various adverse effects, including headache, dizziness, fatigue, and gastrointestinal symptoms [21]. In severe instances, it may lead to intestinal and cardiovascular diseases such as irritable bowel syndrome (IBS) and inflammatory bowel disease (IBD), significantly impacting the health and productivity of individuals in plateau regions [22]. Non-

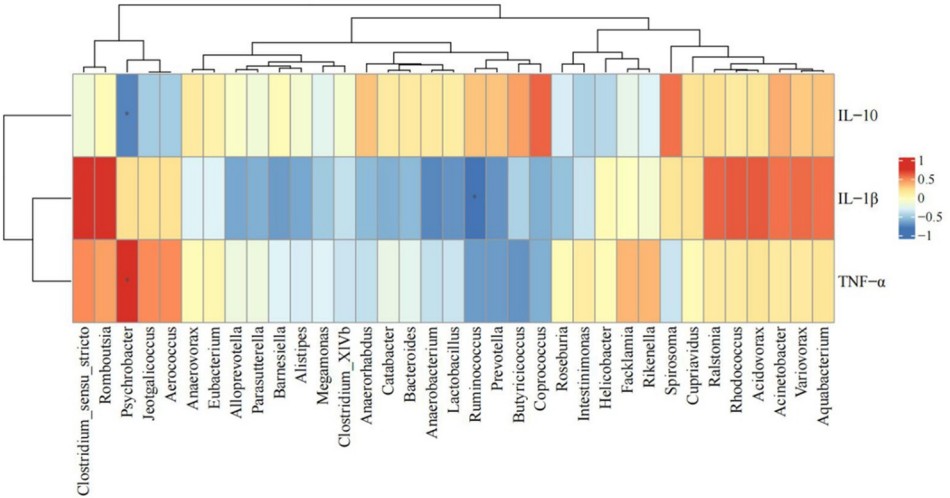

**Fig 9. Correlation analysis between differential gut microbiota and jejunal immune factors in rats.**

steroidal anti-inflammatory drugs (NSAIDs) are commonly used for their anti-inflammatory and analgesic effects in treating inflammatory and cardiovascular conditions [23]. However, prolonged NSAID use can damage the intestinal tract, with potential exacerbation under plateau hypoxia. Resveratrol is a typical polyphenolic compound with preventive and therapeutic effects against many intestinal and cardiovascular diseases [19, 24]. Therefore, the main objective of this study was to investigate the safety of NSAID administration under plateau hypoxia conditions and the modulating effect of resveratrol on intestinal damage induced by the administration of NSAIDs under plateau hypoxia conditions.

In this study, we analyzed the pathological changes of jejunal tissues in SD rats under various treatment conditions. It was observed that the plateau hypoxic environment induced damage to jejunal tissues, characterized by villous atrophy, among other changes. Administration of aspirin via gavage under plateau hypoxia exacerbated intestinal damage, leading to villous shedding and the presence of multiple hemorrhagic spots. These findings indicate that NSAID-based therapies may worsen the inflammatory damage induced by plateau hypoxia. Following intervention with RSV, a significant reduction in intestinal damage was observed, accompanied by the restoration of villous morphology to normal levels.

In addition, previous studies have demonstrated that acute hypoxic exposure can induce alterations in intestinal barrier permeability and the composition of intestinal flora [25, 26]. Tight junction proteins, such as Occludin, Claudin, and ZO-1, are widely recognized as key indicators for assessing intestinal barrier integrity, with changes in their expression reflecting alterations in intestinal permeability [27]. Consistent with earlier findings, our study confirmed that exposure to plateau hypoxic conditions decreased the expression of intestinal tight junction proteins ZO-1 and Occludin [28]. Furthermore, we observed that administration of aspirin under plateau hypoxic conditions further suppressed the expression of ZO-1 and Occludin proteins, indicating that aspirin exacerbates disruption of the intestinal mucosal barrier, corroborating the histopathological observations of jejunal sections. Similarly, following intervention with RSV, the expression of intestinal mucosal barrier proteins was upregulated, suggesting that RSV could ameliorate the damage to the intestinal mucosal barrier caused by NSAIDs. Previous research has also demonstrated that RSV can enhance intestinal barrier integrity by upregulating the expression of proteins involved in maintaining tight junctions between intestinal cells [29]. Assessment of myeloperoxidase (MPO) and superoxide dismutase (SOD), markers of oxidative stress, revealed alterations induced by both the plateau hypoxic environment and aspirin treatment. Myeloperoxidase serves as an objective indicator of neutrophil recruitment, and evaluating MPO activity provides insight into neutrophil infiltration in jejunal tissues [30]. Meanwhile, superoxide dismutase plays a crucial role in maintaining intracellular superoxide radical levels, thereby contributing to cellular homeostasis and health [31, 32]. Our study demonstrated that RSV intervention upregulated SOD expression and downregulated MPO expression, indicating that RSV intervention may mitigate intestinal inflammation by reducing oxidative stress, possibly attributable to its anti-inflammatory and antioxidant properties.

NF-κB, an essential nuclear transcriptional regulator, plays a pivotal role in modulating various cellular processes, including inflammation, immune response, cell proliferation, transformation, apoptosis, tumorigenesis, and others [33]. Dysregulation of NF-κB transcription has been associated with chronic inflammation and cellular apoptosis [10]. Studies have demonstrated that the down-regulation of TLR4 and NF-κB expression mitigates the organism's inflammatory response [34], showing therapeutic efficacy in a murine model of inflammatory bowel disease through the inhibition of the TLR4/NF-κB signaling pathway [28, 35]. To delve deeper into the mechanism underlying RSV-mediated mitigation of intestinal injury, we assessed the expression levels of TLR4/NF-κB/IκB and inflammatory mediators such as IL-1β,

TNF-α, and IL-10. Significant upregulation of TLR4/NF-κB pathway protein expression was observed in rats administered Aspirin. Administering Aspirin in a plateau hypoxic environment led to increased expression of TLR4/NF-κB pathway proteins. RSV intervention resulted in the downregulation of TLR4/NF-κB protein expression. Within the RSVM group, TLR4/NF-κB/IκB pathway protein exhibited the most significant downregulation compared to the other two groups, whereas the IκB protein displayed an opposite trend. Subsequent to RSV intervention, a decrease in the expression of pro-inflammatory cytokines IL-1β and TNF-α was observed, while an increase in the expression of the anti-inflammatory factor IL-10 was noted. These findings indicate that RSV potentially mitigates Aspirin-induced intestinal inflammation by modulating the TLR4/NF-κB/IκB signaling pathway and the release of pro-inflammatory cytokines IL-1β and TNF-α.

To examine the role of gut flora in this mechanism, changes in the composition of gut flora were determined in various treatment groups using 16S rRNA sequencing. The results indicated that administering aspirin via gavage under plateau hypoxia conditions led to alterations in the species richness and diversity of rat intestinal flora. At the phylum level, the HAsp group exhibited higher abundance of *Firmicutes*, lower abundance of *Bacteroidetes*, increased abundance of *Actinobacteria*, and an elevated *Firmicute/Bacteroidetes* ratio compared to the HCk group; the elevated F/B ratio is commonly considered indicative of intestinal flora dysbiosis [36]. Previous studies have also noted dysbiosis of intestinal flora in rats under plateau hypoxic conditions [37, 38], a finding corroborated by our study, which additionally revealed that aspirin gavage under plateau hypoxia exacerbated this dysbiosis. After intervention by RSV, we found no significant change in the composition of gut flora at the portal level in the RSVL and RSVH groups compared to the HAsp group. Whereas, the RSVM group showed 93.97% abundance of Firmicutes compared to the HAsp group, indicating that the RSVM group intervention increased the abundance of intestinal Firmicutes. Whereas, in earlier studies, it was found that RSV intervention could alleviate obesity by decreasing the ratio of F/B in the intestine [39]. However, in this study, we found that RSV intervention upregulated the abundance of Firmicutes in the gut, which may be related to the elevated F/B ratio due to the hypoxic environment of the plateau and Aspirin treatment.

At the genus level, we identified a total of 35 genera exhibiting significant differences. Specifically, six genera showed significant differences ($P < 0.05$) between the HCk and Ck groups: *Ruminococcus*, *Facklamia*, *Parasutterella*, *Jeotgalicoccus*, *Coprococcus*, and Psychrobacter. Following Aspirin treatment in the plateau hypoxic environment, five genera—*Facklamia*, *Jeotgalicoccus*, *Roseburia*, *Psychrobacter*, and *Alloprevotella*—exhibited significant differences ($P < 0.05$); notably, the abundance of *Facklamia* was reduced. In addition, it was observed that intervention with medium-dose RSV significantly increased the abundance of *Clostridium_sensu_stricto*, a member of the Thick-walled Bacteria phylum, known for its significant role in the intestinal tract. Studies have demonstrated that certain strains of *Clostridium_sensu_stricto* can produce a significant amount of extracellular polysaccharide substances [40]. These polysaccharide substances have the potential to enhance the integrity and functionality of the intestinal mucosal barrier, inhibit the permeation of harmful substances, and safeguard the intestinal tract against inflammation and other adverse effects [41]. In addition, it has been found that RSV can alleviate intestinal inflammation and oxidative stress by increasing the amount of *Clostridium_sensu_stricto* in the intestine [42]. This may be one of the reasons why the RSVM group significantly reduced the release of inflammatory factors IL-1β and TNF-α.

*Psychrobacter* and *Ruminococcus* were found to exhibit significant correlations with the expression levels of inflammatory factors through correlation analysis. Among them, *Psychrobacter* exhibited a significant negative correlation with the expression level of IL-10 (r = -0.75, $P < 0.05$) and a positive correlation with the expression level of TNF-α (r = 0.80, $P < 0.05$).

*Ruminococcus* showed association with IL-1β (r = -0.82, *P* < 0.05). *Psychrobacter* exhibited a significant increase in abundance in the HCk group, potentially related to the intestinal inflammation caused by the hypoxic environment of the plateau. Previous studies have shown that the relative abundance of *Psychrobacter* in the ileum correlates positively with the alteration of immunoglobulins and cytokines and may contribute to the upregulation of related inflammatory factors, which aligns with our findings [43]. Previous research has demonstrated that the abundance of *Ruminococcus* decreases significantly in a mouse model of ulcerative colitis [44], indicating a potential association between reduced *Ruminococcus* abundance and intestinal inflammation. In our study, we observed that intervention with RSV led to an increase in the abundance of *Ruminococcus*, potentially associated with the ability of RSV to alleviate intestinal inflammation.

However, this study only simulated the hypoxic plateau environment using a high-pressure, low-oxygen chamber and did not replicate other characteristics of the plateau, such as strong radiation and prolonged sunshine, thereby limiting its ability to fully reflect the effects of the plateau environment on intestinal damage and microorganisms. Moreover, the composition of rat intestinal microorganisms differs from that of humans, and rat experiments do not completely replicate the human intestinal environment. Lastly, the chosen resveratrol dose gradient in this study may not represent the optimal dose for alleviating intestinal injury in rats. Therefore, further studies are required to investigate the causes of intestinal injury in the hypoxic plateau region and to elucidate the specific mechanisms by which resveratrol alleviates intestinal damage.

## 6. Conclusion

In this study, our research found that the administration of NSAIDs drugs under the conditions of plateau hypoxia induced intestinal inflammation and exacerbated intestinal damage. Conversely, resveratrol can attenuate intestinal injury and protect the integrity of the intestinal mucosal barrier by regulating the levels of intestinal mucosal barrier proteins as well as mitigating oxidative stress. Moreover, resveratrol could alleviate intestinal inflammation caused by the administration of NSAIDs-like drugs in rats under plateau hypoxia by regulating the expression of TLR4/NF-κB/IκB pathway proteins and thereby decreasing the release of pro-inflammatory cytokines IL-1β and TNF-α, while simultaneously increasing the release of IL-10. Additionally, our results also revealed that resveratrol could alleviate the intestinal damage caused by NSAIDs drugs in rats by regulating the composition of intestinal flora. The results of the present study may provide effective guidance for the prevention of intestinal disorders caused by the administration of NSAIDs-like drugs in individuals exposed to highland hypoxic conditions.

## Supporting information

**S1 Table. Reuter's scale.**
(DOCX)

**S2 Table. Chiu 's scale.**
(DOCX)

**S1 Fig. PCoA analysis results of different treatment groups.** (A) PCoA analysis of different treatment groups; (B) HAsp vs RSVL; (C) HAsp vs RSVM; (D)HAsp vs RSVH.
(TIF)

**S1 Raw images.**
(PDF)

**S1 Data. Raw measurement data.**
(XLSX)

**S2 Data. ELISA data.**
(XLSX)

## Author Contributions

**Conceptualization:** Shenglong Xue, Wenhui Shi, Ailifeire Tuerxuntayi, Paziliya Abulaiti, Feng Gao.

**Data curation:** Shenglong Xue, Wenhui Shi.

**Formal analysis:** Shenglong Xue, Wenhui Shi, Ailifeire Tuerxuntayi, Paziliya Abulaiti, Zhuoshuyi Liu, Najimangu Remutula, Kailibinuer Nuermaimaiti, Yingying Xing, Kudelaiti Abdukelimu, Weidong Liu.

**Methodology:** Shenglong Xue, Wenhui Shi, Zhuoshuyi Liu, Najimangu Remutula, Kailibinuer Nuermaimaiti, Yingying Xing.

**Project administration:** Feng Gao.

**Resources:** Feng Gao.

**Software:** Shenglong Xue, Wenhui Shi, Yingying Xing, Weidong Liu.

**Supervision:** Tian Shi, Feng Gao.

**Validation:** Shenglong Xue, Tian Shi, Feng Gao.

**Writing – original draft:** Shenglong Xue.

**Writing – review & editing:** Tian Shi, Feng Gao.

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
