## [Decision Letter · Decision Letter 0]

17 Apr 2024

PONE-D-24-11983Resveratrol attenuates non-steroidal anti-inflammatory drug-induced intestinal injury in rats in a high-altitude hypoxic environment by modulating the TLR4/NFκB/IκB pathway and gut microbiota compositionPLOS ONE

Dear Dr. Gao,

Thank you for submitting your manuscript to PLOS ONE. After careful consideration, we feel that it has merit but does not fully meet PLOS ONE’s publication criteria as it currently stands. Therefore, we invite you to submit a revised version of the manuscript that addresses the points raised during the review process.

We look forward to receiving your revised manuscript.

Kind regards,

Palash Mandal

Academic Editor

PLOS ONE

Journal Requirements:

3. PLOS requires an ORCID iD for the corresponding author in Editorial Manager on papers submitted after December 6th, 2016. Please ensure that you have an ORCID iD and that it is validated in Editorial Manager. To do this, go to ‘Update my Information’ (in the upper left-hand corner of the main menu), and click on the Fetch/Validate link next to the ORCID field. This will take you to the ORCID site and allow you to create a new iD or authenticate a pre-existing iD in Editorial Manager. Please see the following video for instructions on linking an ORCID iD to your Editorial Manager account: https://www.youtube.com/watch?v=_xcclfuvtxQ.

In your cover letter, please note whether your blot/gel image data are in Supporting Information or posted at a public data repository, provide the repository URL if relevant, and provide specific details as to which raw blot/gel images, if any, are not available. Email us at plosone@plos.org if you have any questions

Additional Editor Comments (if provided):

Dear Authors,

Thank you for submitting your manuscript to PLOS ONE. After careful consideration, we feel that it has merit but does not fully meet PLOS ONE publication criteria as it currently stands. The shortcomings of this paper needs to be worked out before it can be considered for publication. Therefore, we invite you to resubmit a revised version of the manuscript that addresses the points raised during the review process.

For your guidance, the reviewers' comments are included below.

Thank you for giving us the opportunity to consider your work.

Specific concerns expressed during peer review were:

Comments from Reviewer 1

Dear Dr. Feng Gao and colleagues,

Your manuscript titled "Resveratrol attenuates non-steroidal anti-inflammatory drug-induced intestinal injury in rats in a high-altitude hypoxic environment by modulating the TLR4/NFκB/IκB pathway and gut microbiota composition" has been reviewed. The study is of significant interest, addressing the protective role of resveratrol against NSAID-induced intestinal damage in a high-altitude setting. However, there are several areas that require major revisions to enhance the clarity, depth, and impact of the manuscript. Please find the detailed comments below.

Introduction and Background (Section 1):

The linkage between NSAID usage and intestinal damage in high-altitude environments is well-noted but could be strengthened with recent literature to underline the relevance and timeliness of the study.

Please expand on the molecular mechanisms of the TLR4/NFκB/IκB pathway as it relates to intestinal inflammation, providing a clearer background for readers less familiar with this area.

Methods (Sections 2.1 to 2.9):

In section 2.3, the experimental design should explicitly mention any ethical considerations related to the handling and treatment of the animals beyond the approval by the ethics committee.

Please provide more details on the blinding process during the experimental assessments to avoid bias, especially during the injury scoring and histopathological examinations (Section 2.4, 2.5).

The statistical methods section (2.9) could benefit from additional details regarding the assumptions checked before the analysis and the rationale behind choosing specific statistical tests.

Results (Section 3):

In section 3.2 and 3.3, please provide the exact p-values rather than just stating statistical significance. This would add to the transparency and reproducibility of the results.

The Western blot images and corresponding densitometry data (Section 3.5) should be included in the supplementary files if not in the main manuscript, for validation of the protein expression data reported.

Discussion (Section 5):

The discussion could be improved by contrasting your findings with existing studies, particularly those that have not observed beneficial effects of resveratrol or have different findings regarding gut microbiota changes under similar conditions.

Please address potential limitations of the study, such as the translation of rat model findings to human conditions and the implications of the doses of resveratrol used.

Figures and Tables:

Ensure that all figures are of high quality and appropriately labeled for clarity. Figure legends should be descriptive enough to be understood independently of the main text.

Tables summarizing the raw data and statistical analysis should be provided to enhance the clarity and depth of the reported findings.

References:

Several references appear outdated. Please update these citations with more current research articles that reflect the latest developments in the field.

Ethical Statement:

The ethical statement should be more detailed, especially regarding the specific steps taken to minimize animal suffering and the rationale behind the number of animals used.

Comments from Reviewer 2

Title

•“TLR4/NFκB/IκB pathway”, if all these pathways are involved then why is “/” used? / indicate one of these three pathways. Please clarify the situation

Abstract:

•Line 3, gastrointestinal tract can be abbreviated as GIT and later use the abbreviation instead of the full form.

•ZO-1, first time use the full form then use the abbreviated form.

•RSV, avoid using abbreviations at the start of sentences.

Background:

•In line 5, use NSAIDs, instead of full from “nonsteroidal anti-inflammatory drugs”. There is no consistency in the article. Please check the whole manuscript.

•Rsv, check this for capitalization, in all other places RSV is used. Be consistent.

•More related information can be added in the introduced to make the mechanism of action of given treatment clearer (Resveratrol)

Methodology:

•Experimental design can be represented in the form of a flow chart, or in tabulated form.

Results:

3.2 HE staining results:

•In 2nd last line “mitigates” should be replaced by a suitable word like alleviates. As it is not suitable to use this word here.

3.4:

•RSV can effectively reduce intestinal damage caused by taking nonsteroidal anti-inflammatory drugs under high-altitude hypoxia conditions”, here rearrange this heading, especially “by taking” can be replaced by intake. And use NSAIDs, instead of the full form. Throughout check such types of things, the main headings and subheadings should not be too lengthy.

4.2 Bacterial diversity:

•signifying significantly, check this repetition doesn’t seem attractive.

4.3Microbial composition:

•Clostridium_sensu_stricto, check if sensu_stricto should be italicized according to scientific nomenclature. Apply the same rule to all bacterial phylum and genera, to check the rule for their italicization as above mentioned (for example Clostridiales_Incertae_sedis_XI, Ruminococcus_callidus, etc)

Discussion:

•MPO, avoid abbreviations at the start of a sentence.

Clarity and Structure:

•The clear mechanism between the high altitude, low oxygen pressure, and intestinal damage is not mentioned in the discussion, or in the introduction section

Overall Impression:

•Need to be grammar-checked, spell spell-checked by an expert who is fluent in English.

Recommendations:

•Need to be revised again strictly, by an expert in the given field.

Reviewers' comments:

Reviewer's Responses to Questions

**Comments to the Author**

1. Is the manuscript technically sound, and do the data support the conclusions?

Reviewer #1: Partly

Reviewer #2: Yes

2. Has the statistical analysis been performed appropriately and rigorously? 

Reviewer #1: No

Reviewer #2: Yes

3. Have the authors made all data underlying the findings in their manuscript fully available?

Reviewer #1: No

Reviewer #2: Yes

4. Is the manuscript presented in an intelligible fashion and written in standard English?

Reviewer #1: Yes

Reviewer #2: Yes

5. Review Comments to the Author

Reviewer #1: Dear Dr. Feng Gao and colleagues,

Your manuscript titled "Resveratrol attenuates non-steroidal anti-inflammatory drug-induced intestinal injury in rats in a high-altitude hypoxic environment by modulating the TLR4/NFκB/IκB pathway and gut microbiota composition" has been reviewed. The study is of significant interest, addressing the protective role of resveratrol against NSAID-induced intestinal damage in a high-altitude setting. However, there are several areas that require major revisions to enhance the clarity, depth, and impact of the manuscript. Please find the detailed comments below.

Introduction and Background (Section 1):

The linkage between NSAID usage and intestinal damage in high-altitude environments is well-noted but could be strengthened with recent literature to underline the relevance and timeliness of the study.

Please expand on the molecular mechanisms of the TLR4/NFκB/IκB pathway as it relates to intestinal inflammation, providing a clearer background for readers less familiar with this area.

Methods (Sections 2.1 to 2.9):

In section 2.3, the experimental design should explicitly mention any ethical considerations related to the handling and treatment of the animals beyond the approval by the ethics committee.

Please provide more details on the blinding process during the experimental assessments to avoid bias, especially during the injury scoring and histopathological examinations (Section 2.4, 2.5).

The statistical methods section (2.9) could benefit from additional details regarding the assumptions checked before the analysis and the rationale behind choosing specific statistical tests.

Results (Section 3):

In section 3.2 and 3.3, please provide the exact p-values rather than just stating statistical significance. This would add to the transparency and reproducibility of the results.

The Western blot images and corresponding densitometry data (Section 3.5) should be included in the supplementary files if not in the main manuscript, for validation of the protein expression data reported.

Discussion (Section 5):

The discussion could be improved by contrasting your findings with existing studies, particularly those that have not observed beneficial effects of resveratrol or have different findings regarding gut microbiota changes under similar conditions.

Please address potential limitations of the study, such as the translation of rat model findings to human conditions and the implications of the doses of resveratrol used.

Figures and Tables:

Ensure that all figures are of high quality and appropriately labeled for clarity. Figure legends should be descriptive enough to be understood independently of the main text.

Tables summarizing the raw data and statistical analysis should be provided to enhance the clarity and depth of the reported findings.

References:

Several references appear outdated. Please update these citations with more current research articles that reflect the latest developments in the field.

Ethical Statement:

The ethical statement should be more detailed, especially regarding the specific steps taken to minimize animal suffering and the rationale behind the number of animals used.

Reviewer #2: Title

“TLR4/NFκB/IκB pathway”, if all these pathways are involved then why is “/” used? / indicate one of these three pathways. Please clarify the situation

Abstract:

Line 3, gastrointestinal tract can be abbreviated as GIT and later use the abbreviation instead of the full form.

ZO-1, first time use the full form then use the abbreviated form.

RSV, avoid using abbreviations at the start of sentences.

Background:

In line 5, use NSAIDs, instead of full from “nonsteroidal anti-inflammatory drugs”. There is no consistency in the article. Please check the whole manuscript.

Rsv, check this for capitalization, in all other places RSV is used. Be consistent.

More related information can be added in the introduced to make the mechanism of action of given treatment clearer (Resveratrol)

Methodology:

Experimental design can be represented in the form of a flow chart, or in tabulated form.

Results:

3.2 HE staining results:

In 2nd last line “mitigates” should be replaced by a suitable word like alleviates. As it is not suitable to use this word here.

3.4:

RSV can effectively reduce intestinal damage caused by taking nonsteroidal anti-inflammatory drugs under high-altitude hypoxia conditions”, here rearrange this heading, especially “by taking” can be replaced by intake. And use NSAIDs, instead of the full form. Throughout check such types of things, the main headings and subheadings should not be too lengthy.

4.2 Bacterial diversity:

signifying significantly, check this repetition doesn’t seem attractive.

4.3Microbial composition:

Clostridium_sensu_stricto, check if sensu_stricto should be italicized according to scientific nomenclature. Apply the same rule to all bacterial phylum and genera, to check the rule for their italicization as above mentioned (for example Clostridiales_Incertae_sedis_XI, Ruminococcus_callidus, etc)

Discussion:

MPO, avoid abbreviations at the start of a sentence.

Clarity and Structure:

The clear mechanism between the high altitude, low oxygen pressure, and intestinal damage is not mentioned in the discussion, or in the introduction section

Overall Impression:

Need to be grammar-checked, spell spell-checked by an expert who is fluent in English.

Recommendations:

Need to be revised again strictly, by an expert in the given field.

6. PLOS authors have the option to publish the peer review history of their article (what does this mean?). If published, this will include your full peer review and any attached files.

Reviewer #1: **Yes: **Dr. Prasad Andhare

Reviewer #2: No

---

## [Author Response · Author response to Decision Letter 0]

7 May 2024

Dear Professors：

Thank you so much for giving us the opportunity to revise the article. We are very grateful to the academic editors for their careful attention to the manuscript and to the reviewers for their constructive and insightful comments.We have carefully addressed and clarified the reviewers' concerns, as well as the article format has been adjusted. The specific contents are listed below.

Journal Requirements:

Response:The article has been revised in the format required by the journal.

2. We note that the grant information you provided in the ‘Funding Information’ and ‘Financial Disclosure’ sections do not match.When you resubmit, please ensure that you provide the correct grant numbers for the awards you received for your study in the ‘Funding Information’ section.

Response:The ‘Funding Information’ and ‘Financial Disclosure’ sections has been reworked and the corresponding grant number has been confirmed to be changed.It is as follows:Funding Information: This research was funded by National Natural Science Foundation of China, grant number 82260116; Natural Science Foundation of Xinjiang Uygur Autonomous Region, grant number ZYYD2022A06.

3. PLOS requires an ORCID iD for the corresponding author in Editorial Manager on papers submitted after December 6th, 2016. Please ensure that you have an ORCID iD and that it is validated in Editorial Manager. 

Response:The ORCID of the corresponding authors have been provided and the ORCID of the first author has been added as requested.

Shenlong Xue:https://orcid.org/0009-0009-0650-8300

Correspondence: Feng Gao :https://orcid.org/0000-0002-3320-5702

Response:Already added the appropriate captions for all the images in the article.(See in Images used in article)

All images were corrected through PACE and the resolution of the images (300dpi) was ensured.

5. Please include captions for your Supporting Information files at the end of your manuscript, and update any in-text citations to match accordingly. 

Response:The title of the supporting information document has been added at the end of the article as required by the journal.

6. PLOS ONE now requires that authors provide the original uncropped and unadjusted images underlying all blot or gel results reported in a submission’s figures or Supporting Information files. 

Response:Raw data for all results in the article as well as raw images have been added to the attached material, and we have added the raw data for the sources of the relevant images in the article as well as the results of the software analysis. In addition, the raw protein/blotting results are also included in the submitted supplementary material, and we have labeled the corresponding bands and marker values for each protein in detail. All raw data and images submitted were allowed to be published with the article.

Reviewer #1:

Major comments

1.Introduction and Background (Section 1):

The linkage between NSAID usage and intestinal damage in high-altitude environments is well-noted but could be strengthened with recent literature to underline the relevance and timeliness of the study.

Please expand on the molecular mechanisms of the TLR4/NFκB/IκB pathway as it relates to intestinal inflammation, providing a clearer background for readers less familiar with this area.

Response:We have added an introduction to the relevance of the TLR4/NFκB/IκB signaling pathway to intestinal inflammation as well as and the current state of research in the background section as requested by the reviewers.

The TLR4/NFκB/IκB signaling pathway mainly involves the activation of NF-κB in intestinal tissues, which leads to the release of pro-inflammatory cytokines, thus causing intestinal inflammation and leading to intestinal injury. In the second paragraph of the background section, by describing the plateau hypoxia environment activates the TLR4/NFκB signaling pathway, which causes intestinal inflammation and leads to intestinal injury. The link between plateau hypoxia and intestinal injury is briefly described. In addition, in the third paragraph of the background section, by describing that resveratrol can alleviate intestinal inflammation by regulating the TLR4/NFκB signaling pathway, the connection between resveratrol, the TLR4/NFκB signaling pathway, and intestinal injury was elaborated.

2.Methods (Sections 2.1 to 2.9):

In section 2.3, the experimental design should explicitly mention any ethical considerations related to the handling and treatment of the animals beyond the approval by the ethics committee.

Response:In section 2.2 of the methodology section of the article, it is mentioned in detail that the article passed the appropriate ethical review and complied with the ethical requirements for animal experimentation. In addition, at the end of section 2.2, the ethical considerations of this study involving animals are re-described. The details are given in section 2.2 of the methodology section.

The description is as follows:This experiment was approved by the Animal Ethics Committee for Research Experiments of Xinjiang Medical University (approval number: KY20230209135). It was also conducted in accordance with the Guidelines for Ethical Review of Animal Welfare (GB/T 35892-2018). All efforts were made to minimize their suffering and all experiments were conducted in accordance with the ARRIVE guidelines.

Please provide more details on the blinding process during the experimental assessments to avoid bias, especially during the injury scoring and histopathological examinations (Section 2.4, 2.5).

Response:This section includes the Tissue Injury Score and the Pathology Injury Score. Detailed criteria for the Tissue Injury Score Reuter score and the Pathology Score Chiu scoring have been added. (See in Supplementary Tables)

For the blinded process of this experimental evaluation, the tissue damage score as well as the pathology score were observed by two professional two pathologists using a blinded method and scored according to the appropriate scoring criteria. Changes have been made in the article.

The statistical methods section (2.9) could benefit from additional details regarding the assumptions checked before the analysis and the rationale behind choosing specific statistical tests.

Response:We've reworked and redescribed this section.

Revise as follows:Data were processed using the SPSS 26.0 software. The data were expressed as mean ± standard deviation (SD). Multiple comparisons between the groups were conducted by one-way analysis of variance (ANOVA) followed by the LSD post hoc test.p < 0.05 was considered as significantly different.Graphing was done through Graphpad prism 8.0 software. Univariate and bivariate correlation analysis was performed through R Studio. The * in this study represents comparisons with the plains blank group:*P<0.05, **P<0.01, ***P<0.001, # Comparisons with the plateau NSAIDs-treated group: #P<0.05, ##P<0.01, ###P<0.001.

3.Results (Section 3):

In section 3.2 and 3.3, please provide the exact p-values rather than just stating statistical significance. This would add to the transparency and reproducibility of the results.

Response:Descriptions addressing significance have been added in sections 3.2 and 3.3 of the article. The exact p-values for comparisons between groups of different treatment groups have been added in the Supplementary file.

(See supplementary raw date of the document)

The Western blot images and corresponding densitometry data (Section 3.5) should be included in the supplementary files if not in the main manuscript, for validation of the protein expression data reported.

Response:The original images of the protein blots were submitted in a previous supplement.We resubmitted the original image of the protein blot and the data for the determination of the corresponding density of the protein blot (including gray scale values) in an attachment to this reprint.

(See Westernblot results of the document)

4.Discussion (Section 5):

The discussion could be improved by contrasting your findings with existing studies, particularly those that have not observed beneficial effects of resveratrol or have different findings regarding gut microbiota changes under similar conditions.Please address potential limitations of the study, such as the translation of rat model findings to human conditions and the implications of the doses of resveratrol used.

Response:We have improved the results of the microbiology related discussion section as requested in the discussion section as well. Also added that there are limitations to this study. (See Discussion section of the article)

5.Figures and Tables:

Ensure that all figures are of high quality and appropriately labeled for clarity. Figure legends should be descriptive enough to be understood independently of the main text.

Tables summarizing the raw data and statistical analysis should be provided to enhance the clarity and depth of the reported findings.

Response:We re-uploaded the original high-quality images and added detailed image annotations for the images. (See Images used in article of the document)

In addition, the tables for raw data and statistical analysis were submitted in an earlier attachment, but they existed in a different section of the analysis, so we have consolidated and resubmitted the tables for raw data and statistical analysis, and provided the raw data for the presentation of the images in the statistical analysis, including the Graphpad Prism raw data.

(See in Supplementary raw measurement data and ELISA data).

6.References:

Several references appear outdated. Please update these citations with more current research articles that reflect the latest developments in the field.

Response:We have reworked and replaced some new references in this research area.(See references section of the article)

Reviewer #2:

Major comments

1. “TLR4/NFκB/IκB pathway”, if all these pathways are involved then why is “/” used? / indicate one of these three pathways. Please clarify the situation.

Response:This study involves a pathway consisting of three proteins, TLR4/NFκB/IκB, and “/” is used because similar articles have used “/” to separate different proteins involved in a pathway.

Similar article titles: 

Samaha MM, Nour OA, Sewilam HM, El-Kashef DH. Diacerein mitigates adenine-induced chronic kidney disease in rats: 

Focus on TLR4/MYD88/TRAF6/NF-κB pathway. Life Sci.2023;331:122080. doi:10.1016/j.lfs.2023.12208

2.Abstract:

Line 3, gastrointestinal tract can be abbreviated as GIT and later use the abbreviation instead of the full form.

ZO-1, first time use the full form then use the abbreviated form.

RSV, avoid using abbreviations at the start of sentences.

Response:The gastrointestinal tract has been abbreviated to GIT as requested by the reviewer, and the same has been checked throughout the text, all substituting GIT for gastrointestinal tract.

The use of ZO-1 at the beginning has been changed to Zona Occludens 1 (ZO-1). The entire text has also been checked, and all abbreviations used at the beginning of sentences have been changed to full names. Including RSVs, the use of RSVs at the beginning of sentences has been changed to Resveratrol.

3.Background:

In line 5, use NSAIDs, instead of full from “nonsteroidal anti-inflammatory drugs”. There is no consistency in the article. Please check the whole manuscript.

Response:Non-steroidal anti-inflammatory drugs (NSAIDs), which have been examined throughout the manuscript, are denoted by the acronym NSAIDs, except for the first occurrence, which uses the full term “nonsteroidal anti-inflammatory drugs”

Rsv, check this for capitalization, in all other places RSV is used. Be consistent.

Response:RSV, replacing resveratrol with the acronym RSV, has been checked throughout the manuscript, replacing resveratrol with RSV in all cases

More related information can be added in the introduced to make the mechanism of action of given treatment clearer (Resveratrol)

Response:The introduction section has been reworked and revised to add details about the mechanism of therapeutic action of RSV on the gut. See the introduction section of the article for details.

4.Methodology:

Experimental design can be represented in the form of a flow chart, or in tabulated form.

Response:We have presented the design ideas of this study in the form of a flowchart (see in Flowchart of experimental design of the attachment), but we believe that the flowchart may not fully depict the detailed details related to this study, so we still want to present it in the methodology experimental design section in the same form as it was originally presented.

Results:

3.2 HE staining results:

In 2nd last line “mitigates” should be replaced by a suitable word like alleviates. As it is not suitable to use this word here.

Response:The word “mitigates” has been replaced with “alleviates” .

3.4:RSV can effectively reduce intestinal damage caused by taking nonsteroidal anti-inflammatory drugs under high-altitude hypoxia conditions”, here rearrange this heading, especially “by taking” can be replaced by intake. And use NSAIDs, instead of the full form. Throughout check such types of things, the main headings and subheadings should not be too lengthy.

Response:Regarding this subsection, it focuses on the ELISA results. In response to the reviewer's reference to the title being too long, the title of the paragraph 3.4 has been changed to “Results of Elisa” as requested.

4.2 Bacterial diversity:

signifying significantly, check this repetition doesn’t seem attractive.

Response:The reference to significance in this paragraph has been reworded to reduce the repetition of the reference to significance.

4.3Microbial composition:

Clostridium_sensu_stricto, check if sensu_stricto should be italicized according to scientific nomenclature. Apply the same rule to all bacterial phylum and genera, to check the rule for their italicization as above mentioned (for example Clostridiales_Incertae_sedis_XI, Ruminococcus_callidus, etc)

Response:A review of the relevant literature and the nomenclature of bacteria such as Clostridium_sensu_stricto in the literature revealed that the nomenclature of sensu_stricto in almost all of the literature uses italics (for example Clostridium_sensu_stricto).

At the same time, we checked all bacterial phyla and genera in the text to name all bacterial phyla and genera in the text according to the correct nomenclature (all using italics).

5.Discussion:

MPO, avoid abbreviations at the start of a sentence.

Response:The MPO used at the beginning of the article has been rewritten to the full name Myeloperoxidase, as requested.The entire text has also been checked and the abbreviations at the beginning have been changed to the full name.

Clarity and Structure:

The clear mechanism between the high altitude, low oxygen pressure, and intestinal damage is not mentioned in the discussion, or in the introduction section

Response:Regarding the clear mechanism between high altitude, low oxygen pressure, and intestinal injury, an explanation of the relevant mechanisms of intestinal injury caused by high-altitude hypoxic environments has been added to the background section as requested by the reviewer.

Plateau hypoxia environment can lead to intestinal damage in two ways. First, plateau hypoxia can lead to intestinal flora disorders, which can cause intestinal damage. Secondly, plateau hypoxia will activate the TLR4/NF-κB signaling pathway, leading to the activation of NF-κB, which will cause the release of pro-inflammatory factors, resulting in intestinal inflammation, destroying the intestinal mucosal barrier, and leading to intestinal injury.

---

## [Decision Letter · Decision Letter 1]

28 May 2024

Resveratrol attenuates non-steroidal anti-inflammatory drug-induced intestinal injury in rats in a high-altitude hypoxic environment by modulating the TLR4/NFκB/IκB pathway and gut microbiota composition

PONE-D-24-11983R1

Dear Dr. Feng Gao,

We’re pleased to inform you that your manuscript has been judged scientifically suitable for publication and will be formally accepted for publication once it meets all outstanding technical requirements.

Kind regards,

Palash Mandal

Academic Editor

PLOS ONE

---

## [Editor Report · Acceptance letter]

31 May 2024

PONE-D-24-11983R1 

PLOS ONE

Dear Dr. Gao, 

I'm pleased to inform you that your manuscript has been deemed suitable for publication in PLOS ONE. Congratulations! Your manuscript is now being handed over to our production team.

Kind regards, 

on behalf of

Prof. Palash Mandal 

Academic Editor

PLOS ONE